# Family-led post-ICU discharge intervention for tracheostomized patients in India: Feasibility and formative impact evaluation

Swagata Tripathy[1]*, Asha P. Shetty[2], Upendra Hansda[3], Nanda KumarP.[2],
Alok Kumar Sahoo[1], Mahalingam V.[2], Sujata Mahapatra[2], Jayanta Kumar Mitra[1],
Parnandi Bhaskar Rao[1], Kasturi Sanyal[1], Itimayee Panda[1], Guruprasad N.[4],
Jagannath Sahoo[5], Rashan Haniffa[6,7,8,9], Abi Beane[6,10]

1 Department of Anesthesia and Critical Care, All India Institute of Medical Sciences Bhubaneswar,
Bhubaneswar, Odisha, India, 2 College of Nursing, All India Institute of Medical Sciences Bhubaneswar,
Bhubaneswar, Odisha, India, 3 Department of Trauma and Emergency, All India Institute of Medical
Sciences Bhubaneswar, Bhubaneswar, Odisha, India, 4 Department of Neurosurgery, All India Institute
of Medical Sciences Bhubaneswar, Bhubaneswar, Odisha, India, 5 Department of Physical Medicine and
Rehabilitation, All India Institute of Medical Sciences Bhubaneswar, Bhubaneswar, Odisha, India, 6 Centre
for Inflammation Research, University of Edinburgh, Edinburgh, Scotland, United Kingdom, 7 Critical
Care Medicine, University College London, London, United Kingdom, 8 National Institute of Critical Care
Medicine and Training, Colombo, Sri Lanka, 9 Mahidol Oxford Tropical Medicine Research Unit, Bangkok,
Thailand, 10 Institute for Regeneration and Repair, The University of Edinburgh, Edinburgh, Scotland,
United Kingdom

* tripathyswagata@gmail.com

journal.pone.0348345

of Science, UNITED KINGDOM OF GREAT
BRITAIN AND NORTHERN IRELAND

**Peer Review History:** PLOS recognizes the
benefits of transparency in the peer review
process; therefore, we enable the publication
of all of the content of peer review and
author responses alongside final, published
articles. The editorial history of this article is
available here: https://doi.org/10.1371/journal.
pone.0348345

## Abstract

### Background

Survival after critical illness is increasing, but many patients remain chronically
critically ill (CCI), dependent on tracheostomy and ongoing care. In low- and middle-
income countries (LMICs), long-term facilities are scarce, costly, and often excluded
from insurance, leading to prolonged ICU stays, hospital-acquired complications, and
constrained bed capacity. Family-centred discharge interventions may provide a safe,
cost-conscious alternative, but evidence on feasibility, acceptability, and implementa-
tion success in LMICs is limited.

### Methods

We conducted a mixed-methods formative evaluation of the AIIMS ICU Rehabili-
tation (AIR) intervention, a co-designed, multi-component programme supporting
the transition of tracheostomised patients from ICU to home care at a public tertiary
hospital in India (2021–2024). The intervention comprised structured carer training, a
mobile health communication platform, an equipment rental-retrieval bank, and post-
discharge follow-up including home visits. Implementation outcomes were assessed
using the Medical Research Council framework for complex interventions and the
RE-AIM (Reach, Effectiveness, Adoption, Implementation, Maintenance) framework.

**Data availability statement:** All relevant data are within the paper and its Supporting Information files.

**Funding:** National Health Mission India, Wellcome Trust, Indian Council of Medical Research (ICMR).

**Competing interests:** The authors have declared that no competing interests exist.

**Abbreviations:** AIR, AIIMS ICU Rehabilitation; ICU, Intensive Care Unit; CCI, Chronically Critically Ill; mHealth, Mobile Health; RT, Research Team; TT, Treating Team; IP, Intervention Provider; AIM, Acceptability of Intervention Measure; IAM, Intervention Appropriateness Measure; FIM, Feasibility of Intervention Measure; CFIR, Consolidated Framework for Implementation Research; RE-AIM, Reach, Effectiveness, Adoption, Implementation, and Maintenance; MRC, Medical Research Council; CReDECI 2, Criteria for Reporting the Development and Evaluation of Complex Interventions (Revised Guideline); EQ-5D, EuroQOL 5 Dimensions; CBS, Caregiver Burden Scale; LMIC, Low- and Middle-Income Countries; HCW, Healthcare Worker.

Quantitative measures included validated implementation scales assessing acceptability, feasibility and appropriateness of the intervention, as well as carer confidence, quality of life and caregiver burden. Semi-structured interviews assessed stakeholder barriers and facilitations to implementation analysed using the Consolidated Framework for Implementation Research (CFIR).

## Results

Of 762 patients screened, 314 were eligible and 300 dyads (96%) consented. Recruitment shifted from research-led to 98.5% clinician or family referral by year three. Carers and patients rated the intervention highly feasible, acceptable, and appropriate (median AIM 20, IAM 19.5, FIM 19.5), with greater endorsement than healthcare staff. Confidence improved with training: 66% of carers completed at least three structured training sessions and 61% achieved predefined competence after three training sessions. The mobile application was installed by 74% of dyads although WhatsApp was frequently preferred for communication with the care team. More than half accessed equipment through the rental–retrieval bank, and 91% of eligible families received in-person post-discharge follow-up. Qualitative findings identified barriers including carer reluctance in younger trauma cases, medicolegal concerns, fragmented training, and socioeconomic constraints. Facilitators included trust in clinicians, flexible training approaches, and ongoing post-discharge support.

## Conclusion

The AIR intervention is feasible, acceptable, and adaptable in a public LMIC setting. Carer confidence increased during the intervention period and family-led home transition for tracheostomised ICU survivors was possible while identifying contextual barriers and facilitators relevant for scale-up. These findings informed refinement of the intervention, such as including targeted patient selection, early recruitment, and peer-supported training, and will guide a planned multicentre summative evaluation assessing effectiveness, sustainability, and cost-effectiveness.

## Introduction

Advances in critical care have improved survival after severe infection or injury, but many survivors face long-term dependency due to altered consciousness, ongoing physical care needs, or reliance on therapies such as tracheostomy and mechanical ventilation [1,2] These patients, often described as chronically critically ill (CCI), frequently require prolonged hospitalisation or long-term residential care, including 24-hour tracheostomy management, ventilatory support, and rehabilitation [3].

In resource-limited settings, dedicated long-term care facilities are scarce or absent. Where available, they are often unaffordable, excluded from health insurance coverage, and can drive catastrophic health expenditure, destabilising families economically [4,5] Consequently, CCI patients may remain in ICUs or acute hospital

beds, exposing them to nosocomial complications while limiting bed availability for new critically ill patients in already overstretched systems [2].

Family-centred interventions that enable timely, safe discharge from hospital to home offer a potential solution [6]. These programmes typically shift aspects of care from clinical teams to family members (or lay carers chosen by families), supporting home-based management while easing pressure on hospital services. Benefits may include reduced healthcare expenditure, improved ICU throughput, and opportunities for patients to be cared for in familiar environments.

Such interventions are inherently multifaceted and require collaboration between hospital and community stakeholders. They often include training in tracheostomy care, nutrition, personal hygiene, and rehabilitation. Emerging evidence from low- and middle-income countries (LMICs), particularly in stroke and cancer care, suggests that family-centred discharge interventions are feasible and acceptable [7–14]. Their potential impact may be even greater in the critical care context, where ICU bed capacity is limited and social structures, such as multi-generational households, can support care at home. However, most existing studies are observational, small-scale, and lack detailed descriptions of intervention design, implementation strategies, and contextual factors influencing outcomes, including feasibility, acceptability, stakeholder buy-in, and resource availability (e.g., internet, technology, transport).

We therefore conducted a formative impact evaluation of the AIIMS ICU Rehabilitation (AIR) intervention, a family-centred discharge programme for tracheostomised patients [15]. The objectives were to assess intervention feasibility, implementation reach, identify barriers and facilitators to implementation, and evaluate its formative impact on clinical and patient-carer dyad-centred outcomes. These findings will guide refinement of the intervention and inform a planned multicentre summative implementation evaluation, which will formally assess intervention effectiveness, process outcomes, patient quality of life, and caregiver burden (S1 File).

## Methods

### Design

We employed a mixed-methods design following the Medical Research Council (MRC) guidelines for evaluating complex healthcare interventions [16]. This evaluation combined self- and researcher-administered psychometric assessments of intervention feasibility, acceptability, appropriateness, and behaviour change (confidence in skills and carer burden), together with implementation outcomes of reach, fidelity and adoption. Semi-structured interviews were conducted to elicit stakeholders' perceived barriers and facilitators to intervention and implementation. Additionally, prospective data were captured on patient–carer dyad characteristics, patient safety, and clinical outcomes.

Consistent with MRC recommendations, the RE-AIM framework (Reach, Effectiveness, Adoption, Implementation, Maintenance) and the MRC-CReDECI 2 tool guided the assessment of implementation outcomes used during this formative evaluation, which will inform the summative multicentre AIR intervention [17,18]. The Consolidated Framework for Implementation Research (CFIR) informed the analysis and interpretation of barriers and facilitators [19].

### Intervention details

A detailed description of the AIR intervention has already been published [15]. Briefly, AIR aims to facilitate the safe transition of tracheostomised patients from ICU care to family-led home care. It comprises four components:

1. **Hands-on practical training** for family carers, covering tracheostomy management (suctioning, cleaning, stoma care), oxygen management, nutrition (including nasogastric tube feeding and blood sugar monitoring for diabetic patients), and bed care (urinary catheter care, personal hygiene, and skin care).

2. **mHealth application** providing practical guidance using training videos/ guides and enabling voice and text communication between carers and the ICU healthcare team, particularly during and after hospital discharge.

3. **Equipment bank** supplying essential items, including home ventilators, hand-held suction devices, oxygen, nebuliser kits, oximeters, hospital beds, and wheelchairs, on a rental–retrieval model free for the first month.

4. **Post-discharge home visits** with at least one visit between discharge and 30, supplemented by telephone follow-ups, offering semi-structured support to families.

### The AIR Research Team

The multidisciplinary team included experts in intensive care, neurosurgery, emergency medicine, physical rehabilitation, nutrition, nursing, clinical psychology, and social science. This team was intended to lead screening, approach, consent, and recruitment in consultation with treating clinicians.

### Setting

The study was conducted in the All-India Institute of Medical Sciences (AIIMS) Bhubaneswar, a tertiary care teaching hospital in Odisha India. AIIMS has approximately 100 ICU beds across multiple units including central (mixed medical-surgical) ICU, neuro-respiratory ICU, neurosurgical ICU, and emergency ICUs. Patients for the intervention were recruited from the neurology, neurosurgical and mixed medical–surgical ICUs. These ICUs have a total capacity of 35 beds and manage critically ill patients requiring advanced monitoring and organ support. The units function with a patient-to-nurse ratio of approximately 1:2–3, facilitating continuous bedside monitoring and intensive nursing care. Clinical management is supervised by trained intensivists who provide 24-hour on-call coverage, along with full-time in-person medical staff available in the unit at all times. Odisha is primarily rural with a rapidly urbanising sector, a population of 42 million, and a significant tribal community [20]. AIIMS Bhubaneswar serves over one million outpatients monthly and provides tertiary neuro, critical care, trauma, and transplant services for neighbouring states [21].

### Eligibility and recruitment

Eligible patients were ICU-admitted adults requiring tracheostomy with ongoing physical dependency. Daily screening ensured physiological stability (hemodynamically stable and off vasoactive drugs, oxygen <5 L/min, afebrile, no recent or uncontrolled seizures) and the availability of a family member or family chosen lay carer to participate. Patients were excluded if they were <18 years of age, decannulated prior to discharge, or lacked an available caregiver able to partic-ipate in training. Written consent was obtained from the next of kin after provision of relevant study information and an opportunity to discuss any queries with the research team and the lead clinician. Convenience sampling was used for all carer-dyad psychometric assessments, except for purposive interviews. Psychometric assessments were conducted among dyads recruited during the first year of implementation to evaluate early perceptions of the intervention. The rep-resentativeness of the participating dyads and the recruited cohort was described. A dyad refers to the patient and their primary family caregiver participating together in the intervention.

While all eligible patient-carer dyads and stakeholders were offered participation in the intervention, convenience sampling was used for psychometric assessments, and purposive sampling was used for qualitative interviews to ensure representation of different stakeholder groups and experiences with the intervention.

### Implementation frameworks and outcome measures

The Medical Research Council (MRC) framework for developing and evaluating complex healthcare interventions guided the overall design and formative evaluation of the intervention, including assessment of feasibility, implementation pro-cesses, and contextual influences on delivery [22]. To assess implementation performance across multiple domains, we applied the RE-AIM framework (Reach, Effectiveness, Adoption, Implementation, Maintenance), which provides a

structured approach for evaluating public health interventions across individual and organisational levels [17]. Both are widely used and accepted guides for implementation research internationally. Stakeholder perceptions of the intervention were assessed using validated implementation outcome measures: the Acceptability of Intervention Measure (AIM), Intervention Appropriateness Measure (IAM), and Feasibility of Intervention Measure (FIM) [23]. Additional quantitative measures included carer confidence scores related to tracheostomy and home-care skills, as well as patient and caregiver outcome measures including quality-of-life [24] and caregiver burden assessments [25]. To explore contextual factors influencing implementation, semi-structured qualitative interviews were conducted with carers and healthcare workers. Qualitative data were analysed using the Consolidated Framework for Implementation Research (CFIR), which provides a structured taxonomy for identifying barriers and facilitators across domains including intervention characteristics, inner and outer settings, individual characteristics, and implementation processes; and is again widely accepted and used in international implementation research [19].

Formative evaluation assessments

**Feasibility, acceptability, and appropriateness.** Stakeholder perceptions were assessed using the Acceptability of Intervention Measure (AIM), Intervention Appropriateness Measure (IAM), and Feasibility of Intervention Measure (FIM) in dyads recruited during the first year of the study. Each tool uses a four-point Likert scale (3–4 indicating greater feasibility, acceptability, or appropriateness) [23,26]. Tools were translated into Odia and Bengali, linguistically validated via the Brislin method and administered by the research team [27].

**Reach, fidelity and adoption.** Selected RE-AIM domains (Reach, Fidelity, Adoption) were assessed at consent, training, and up to 30 days post-discharge [17]. Reach was measured by the proportion of eligible patients consenting, demographics, and reasons for non-participation. Referral sources (research team, clinicians, families) were documented and used to assess the potential for adopting the intervention. Additional fidelity measures included: dyads completing ≥3 training sessions; mobile device availability; equipment access, rental, or purchase; and completion of home visits. Deviations from intended intervention, such as equipment unavailability, were reported along with adaptations in response to feedback.

**Intervention outcomes.** Carer self-confidence in performing care components was assessed post-training using a 0–10 Likert scale (0 = not confident, 10 = fully confident). A minimum of three hands-on sessions were planned, with additional sessions offered until carers scored ≥7 on each component. Trainers provided an objective confidence assessment, and session duration was recorded.

Patient and carer quality of life and carer burden were measured using the EuroQOL-5D (EQ-5D) and the Caregiver Burden Scale (CBS) at discharge, day 14, and day 28 [24,25]. Clinical outcomes, including ICU-to-hospital discharge time and vital status at discharge, day 14, and day 28, were also recorded as an assessment of feasibility ahead of the future summative evaluation.

**Stakeholder perceived barriers and facilitators.** Semi-structured interviews were conducted with patient–carer dyads at intervention completion (immediately prior to or following discharge home). Purposive sampling ensured representation of decannulated and cannulated patients, as well as survivors and deceased cases. Participants were approached in person, by email, or via WhatsApp, with a target of ≥20 assessments to achieve saturation. CFIR was selected for its suitability for evaluating multi-component interventions involving technology and diverse stakeholders, and informed both interview guides and the subsequent synthesis of experiences [28].

## Data collection and analysis

A simplified timeline of the AIR implementation and evaluation activities is presented in Fig 1, while details are pre-published [15]. Data were collected prospectively from April 2021 to March 2024 via structured electronic case record forms on REDCap [29]. Data included demographics, social characteristics, together with daily data on reach, intervention fidelity, clinical parameters, and outcomes. Descriptive analysis was used to summarise case mix, implementation reach,

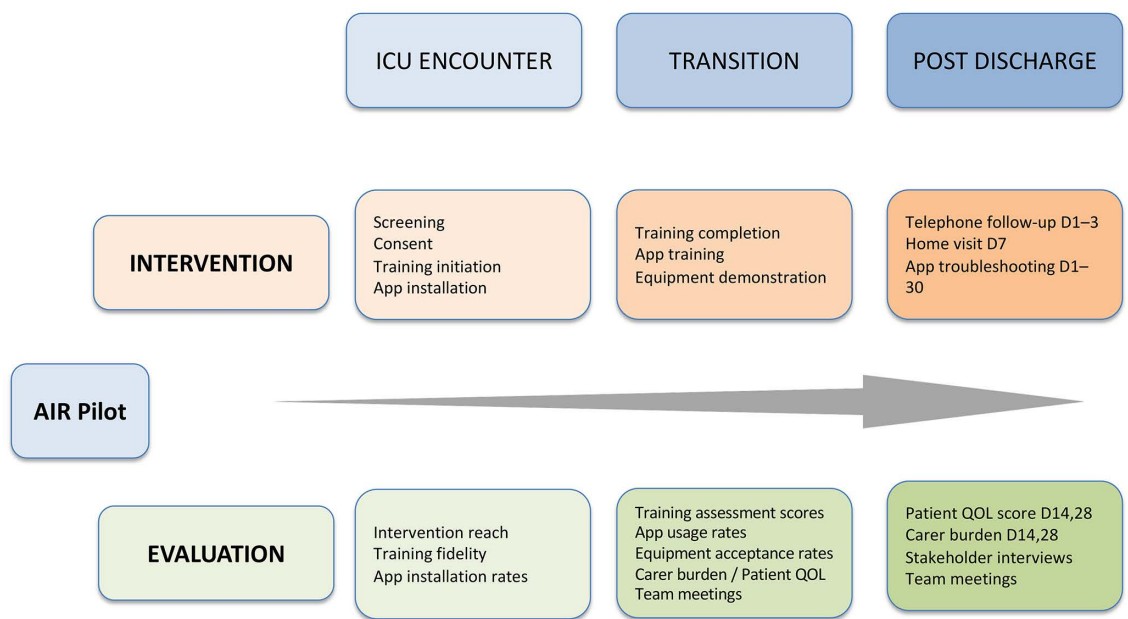

**Fig 1. AIR Implementation Timeline and Measurements.** The figure summarises the stages of the AIR intervention and the corresponding process evaluation measures collected during implementation.

intervention fidelity, and outcomes. AIM, IAM, FIM, and confidence scales were analysed per published methods, stratified by stakeholder group.

Quantitative data were analysed using descriptive statistics. Continuous variables were summarised using means with standard deviations (SD) or medians with interquartile ranges (IQR), depending on distribution, and categorical variables were summarised as frequencies and percentages. Changes in patient quality of life (EQ-5D) and caregiver burden (CBS) across time points were analysed using paired non-parametric tests (Wilcoxon signed-rank or Friedman tests) for repeated measures as appropriate. Effect sizes were estimated using rank-based measures where relevant, and 95% confidence intervals were reported where applicable. All analyses were exploratory and intended to inform future hypothesis-testing studies. Missing data were minimal and primarily related to loss to follow-up among non-survivors. No imputation was performed. Analyses were conducted using available cases for each analysis; therefore, the number of observations varied depending on data availability for the variables included.

The interview guide for the qualitative assessment was prepared by members of the research team and can be found in Supplement 2. Interviews were conducted by project team members familiar with the AIR programme, trained in qualitative methods, but not involved in intervention delivery. Field notes and interview transcripts were reviewed by members of the research team and analysed using an interpretive approach. Analysis was carried out alongside ongoing data collection, allowing emerging issues to be explored in later interviews. Senior researchers with a track record in critical care service improvement and implementation facilitated the reflexive sessions. Emergent findings were discussed with both senior research and the wider multidisciplinary AIR team to reflect on interpretation and incorporate perspectives from stakeholders [15]. Interview transcripts were analysed using inductive thematic analysis. Codes were developed iteratively from the data and subsequently mapped to constructs within the Consolidated Framework for Implementation Research (CFIR). Data organisation and coding were conducted using Microsoft Excel [30].

## Ethics

Institutional ethics committee approval was obtained, and the trial was prospectively registered (CTRI/2020/11/029443). Written informed consent was obtained from all participants.

## Results

A total of 300 patient–carer dyads were recruited over 36 months (Fig 2). Feasibility, implementation reach, and fidelity, together with formative outcomes (confidence, carer burden), and clinical outcomes, are described in the main manuscript and reported in Tables 1 and 2, Fig 4, and S1 Fig. Additional results exploring socio-economic characteristics and reasons for loss of fidelity are reported in S1 Table and S2 Fig. CFIR domains that dominated stakeholders' perceptions of barriers and facilitators are described below, together with illustrative quotes. Expanded CFIR constructs are reported in Fig 3. Intervention and implementation adaptations are summarised in S1 Table.

### Implementation outcomes

**Reach and adoption:** Of 762 dyads screened, 314 were eligible and 300 (96%) consented; 272 (86.1%) at first approach. Hesitancy to participate was usually linked to family discussions about nominating carers. Of the 14 dyads who declined,

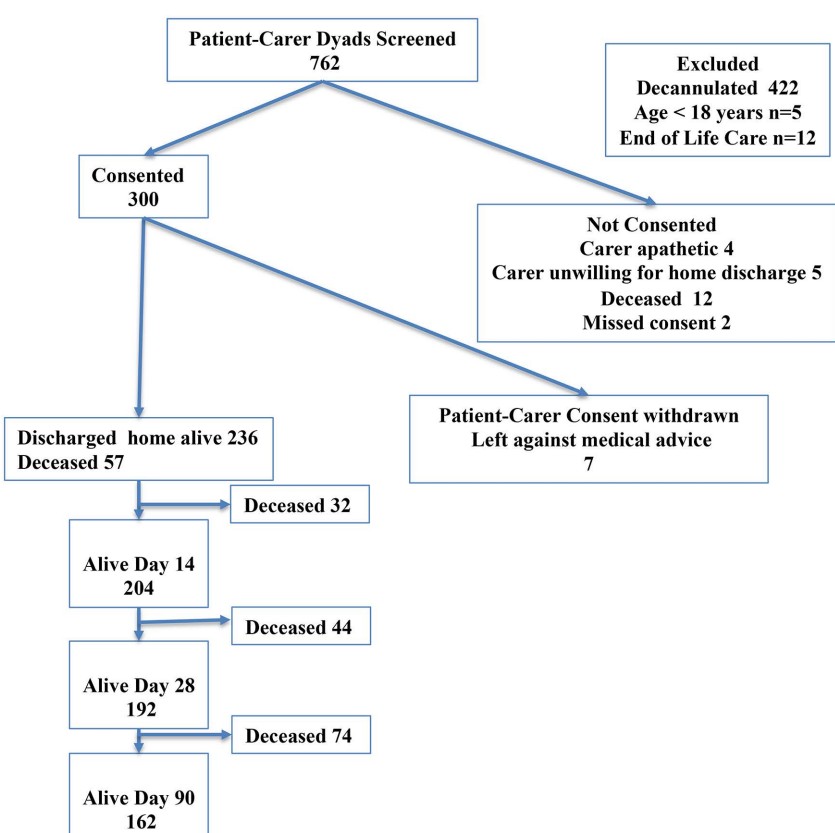

**Fig 2. Screening and Recruitment Flowchart.** Flowchart depicting the screening and recruitment process of patient-carer dyads into the AIR Pilot study, including numbers approached, excluded, and enrolled.

**Table 1.** Demographic and clinical characteristics of Patient-Carer Dyads.

| | All Enrolled Dyads (n = 300) | Dyads participating in Psychometric AIM, IAM, and FIM Assessment (n = 33) |
|---|---|---|
| Patients | | |
| Age (years), Mean (SD) | 50.5 (16.6) | 50.8 (16.4) |
| | | |
| Sex, n (%) Male Female | 216 (72.0%) 84 (28.0%) | 22 (66.7%) 11 (33.3%) |
| | | |
| | | |
| Education Level, n (%) | | |
| Postgraduate | 9 (3.0%) | 0 (0.0%) |
| Undergraduate | 87 (29.0%) | 3 (11.1%) |
| High School | 189 (63.0%) | 26 (78.8%) |
| Illiterate | 15 (5.0%) | 4 (12.1%) |
| | | |
| Occupation, n (%) | | |
| Professionals & Technicians | 35 (11.7%) | 1 (3.0%) |
| Elementary | 141 (47.0%) | 14 (42.4%) |
| Unemployed | 104 (34.7%) | 13 (39.4%) |
| Others | 20 (6.7%) | 5 (15.2%) |
| | | |
| Diagnosis, n (%) | | |
| Post planned neurosurgery (i.e., space-occupying lesions, including malignancy, neuro-vascular malformation) | 76 (25.3%) | 9 (27.3%) |
| Traumatic Brain Injury (TBI) | 44 (14.7%) | 7 (21.2%) |
| Stroke | 65 (21.7%) | 7 (21.2%) |
| Primary cerebral infection (i.e., viral, parasitic and bacterial) | 24 (8.0%) | 5 (15.2%) |
| Chronic Lung Disease (i.e., acute exacerbation for infection and non-infectious) | 12 (4.0%) | 3 (9.1%) |
| Non-infectious encephalopathy (such as hypoxic, metabolic) | 74 (24.7%) | 2 (6.1%) |
| Others (degenerative disc disorder, ca lung, neuromuscular disease) | 5 (1.7%) | – |
| | | |
| Clinical Outcomes | | |
| Hospital survival | 236 | 28 |
| Day 14 (post-hospital discharge) survival | 204 | 24 |
| Day 28 (post-hospital discharge) survival | 192 | 24 |
| | | |
| Length of Stay (days), Mean (SD) | 12.0 (8.7) | 11.0 (8.9) |
| | | |
| Carer | | |
| Age (years), Mean (SD) | 34.2 (11.3) | 33.0 (9.7) |
| | | |
| Sex, n (%) Male Female | 175 (58.3%) 125 (41.7%) | 25 (75.8%) 8 (24.2%) |
| | | |

*(Continued)*

**Table 1.** (Continued)

| | All Enrolled Dyads (n = 300) | Dyads participating in Psychometric AIM, IAM, and FIM Assessment (n = 33) |
|---|---|---|
| Education Level, n (%) | | |
| Postgraduate | 17 (5.7%) | 2 (6.1%) |
| Graduate | 146 (48.7%) | 10 (30.3%) |
| High School | 129 (43.0%) | 18 (54.5%) |
| Illiterate | 8 (2.7%) | 1 (3.0%) |
| | | |
| Occupation, n (%) | | |
| Professionals & Technicians | 22 (7.3%) | 3 (9.1%) |
| Elementary | 133 (44.3%) | 15 (45.5%) |
| Unemployed | 122 (40.7%) | 11 (33.3%) |
| Others | 23 (7.7%) | 2 (6.1%) |

**Table 2. AIM, IAM, and FIM scores by stakeholder group.**

| Stakeholders | n | AIM* Median (Range, IQR) | IAM* Median (Range, IQR) | FIM* Median (Range, IQR) |
|---|---|---|---|---|
| **Patient-Carer dyads** | 33 | 20 (15–20, 0) | 19.5 (15–20, 0) | 19.5 (13–20, 0) |
| **Health Care Workers** | 42 | 20 (19-20,0.75) | 20 (19-20 1) | 19 (16 17–1920 4) |
| • Consultants | 14 | 20 (17–20, 0) | 20 (17–20, 0.75) | 20 (12–20, 1.75) |
| • Trainee Doctors | 8 | 19.5 (16–20, 2.5) | 19 (16–20, 2.5) | 16 (13–20, 2.75) |
| • Nurses | 14 | 20 (12–20, 1.75) | 19 (14–20, 1.75) | 19 (13–20, 2.75) |
| • Technical Staff | 6 | 20 (18–20, 0.5) | 20 (16–20, 0) | 19 (10–20, 6.5) |
| **Non-clinical- carer stakeholders** | 11 | 19(10.5-20, 9.5) | 17(10.5-20, 9.5) | 16(8.5-20, 11.5) |
| • mHealth App Developers | 3 | 20 (20–20, 0) | 20 (20–20, 0.5) | 20 (20–20, 1.5) |
| • Equipment Vendors | 2 | 20 (20–20, 0) | 20 (20–20, 0) | 20 (20–20, 0) |
| • Funding Agency | 3 | 18 (18–19, 0.5) | 17 (17–18, 0.5) | 16 (13–17, 1.5) |
| • Administrative Staff | 3 | 17 (16–19, 0.5) | 16 (16–17, 0.5) | 16 (13–16, 1.5) |

Values are presented as median (range, interquartile range [IQR]). Range represents the minimum–maximum values observed. IQR represents the interquartile range (25th–75th percentile).

*AIM = Acceptability of Intervention Measure; IAM = Intervention Appropriateness Measure; FIM = Feasibility of Intervention Measure.

five doubted family-led discharge was possible, and were typically in the context of younger patients with acute illness (including trauma). In the first two years, recruitment was research team–initiated (141/235, 60%), followed by treating team referrals (87, 37%) and carer self-referral (6, 2.6%). In the third year, 65 dyads were recruited, and nearly all referrals were initiated directly by treating teams or families (64/65, 98.5%).

## Intervention outcomes

**Feasibility:** The AIM, IAM, and FIM tools were completed by 86 stakeholders (33 dyads, 42 healthcare workers [HCWs]or persons associated with the project). Demographics are shown in Table 1, scores in Table 2. Carers rated the intervention highly acceptable, appropriate, and feasible (median AIM 20, IAM 19.5, FIM 19.5). Dyads consistently scored higher than

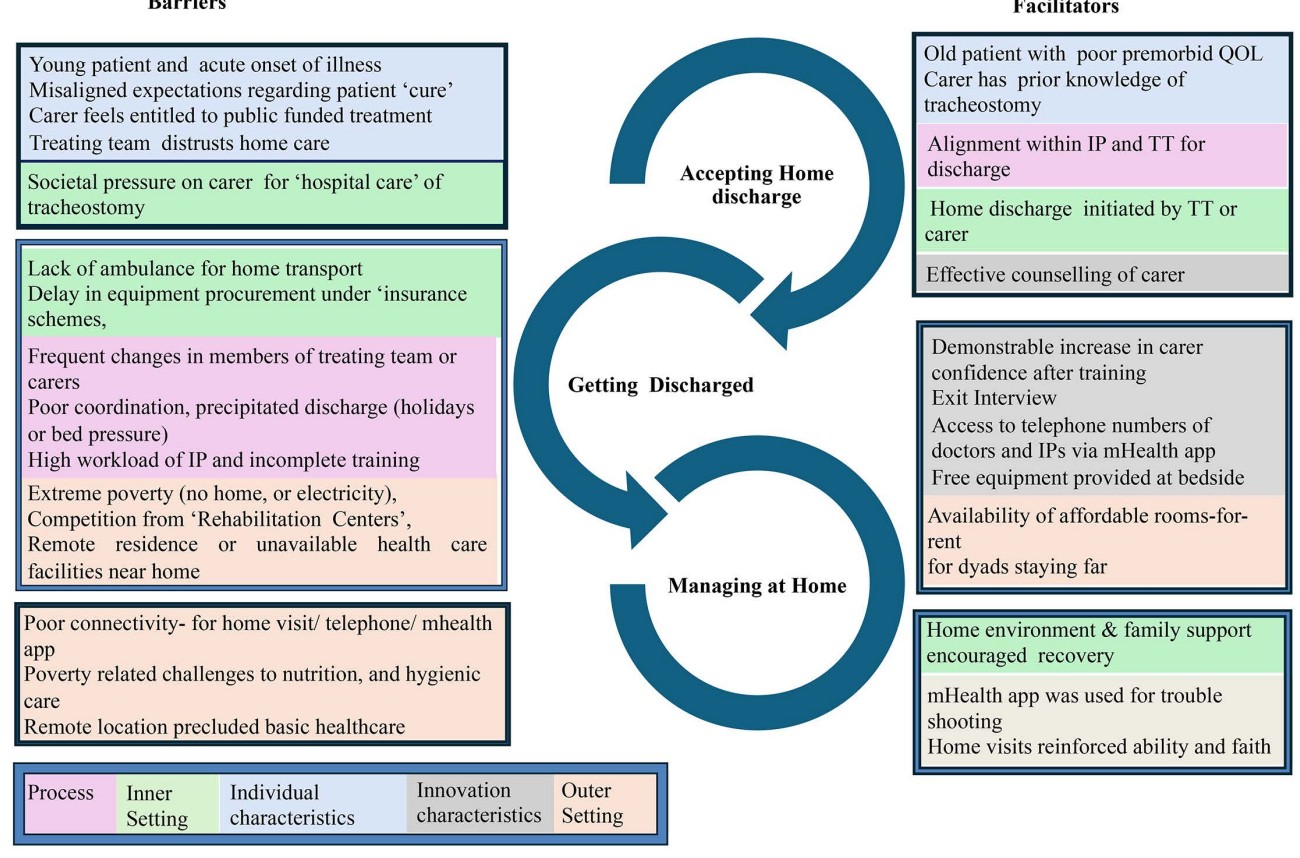

**Fig 3. Emergent Barriers and Facilitators Mapped to the CFIR Framework.** Conceptual mapping of barriers and facilitators impacting implementation, categorised by CFIR domains. Colour-coded per CFIR construct. CFIR = Consolidated Framework for Implementation Research.

HCWs; the lowest feasibility ratings came from HCWs, funders, and administrative staff. Carer education and employment were not associated with scores or confidence scales (S1 Table).

**Confidence, QoL and carer burden:** Carer confidence improved across all training components (Fig 4). Confidence scores >7 was reported by 61% after three sessions, 30% after 4, and 9% after 5. All dyads were followed at discharge, day 14, and day 28. For survivors, median carer-reported patient QOL improved from 52 (IQR 7.5) at discharge, to 58 (IQR 11) at day 14, and 60 (IQR 10) at day 28. The improvement was significant between hospital discharge and day 14 (p < 0.001) and hospital discharge and day 28 (p < 0.001), while the difference between day 14 and day 28 was not statistically significant (p = 0.06).

Carer burden declined over the same period: 62 (IQR 8), 57 (IQR 9), and 53 (IQR10). Reductions were significant between discharge and day 14 (p < 0.001), discharge and day 28 (p < 0.001), and day 14 and day 28 (p = 0.004).

## Fidelity

*Training*: Of 300 carers, 280 (93%) attended at least one session; 200 (66.3%) completed three or more. Sessions shortened from a mean of 42 minutes (SD 18) initially to 30 minutes (SD 12). Of the 100 who did not complete three sessions, 31 were carers of patients who died. Remaining gaps in training were due to patient discharge or transfer in response to bed pressures or to inconsistent attendance by carers (Supplement Figure 2).

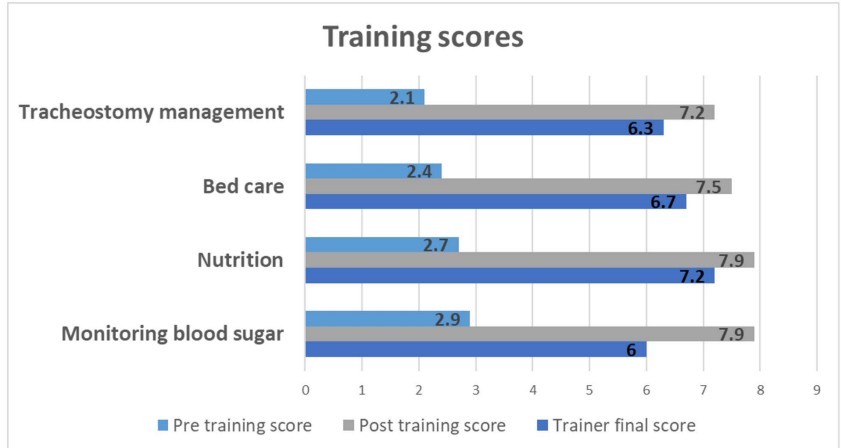

**Fig 4. Training Scores.** Pre -and post-tests are self-scored by carers before and after the final training in each training module. The Trainer score is the score given by a nurse observer from the research team after the carer reaches the best score.

*mHealth app*: Launched in October 2021, the app was available to 266 dyads; 197 (74.1%) installed it across 348 devices. Barriers included a lack of smartphones (21 dyads) and installation errors (11 dyads). The complete installation process took approximately 1 hour. All users accessed the app at least once; training videos and FAQs were the most popular features.

*Equipment bank*: All dyads were offered access. Of these, 166 (55%) borrowed equipment free of charge for one month, and 134 (45%) purchased items via their existing insurance packages. Commonly loaned items were suction machines (129, 78%), pulse oximeters (126, 76%), and nebulisers (108, 65%). Ventilators were loaned to 16 dyads (10%).

*Clinical outcomes and home follow-up*: Median time to discharge was 10 (IQR 3.5) days in years one and two, reducing to 4.4 (IQR2.8) in year three. All dyads (n = 300) received phone follow-up within 1–3 days. Of 154 patients living within 200 km, 140 (91%) received in-person visits.

**Stakeholder perceived barriers and facilitators to implementation**

Interviews were conducted with 14 dyads, 6 HCWs, and two non-clinical stakeholders. Fifteen barriers and twelve facilitators condensed from the CFIR constructs were identified. Four CFIR domains dominated stakeholder perceptions of the likely barriers to, and success of the intervention and its implementation, implementation process, individual characteristics, outer settings and innovation characteristics. These are described below, together with illustrative quotes, and are summarised further in Fig 3.

**Implementation process**: Trust between carers and the research team was central to implementation success. The tailored approach to training, together with the opportunity to repeat sessions, fostered confidence and acceptability for all stakeholders.

Dyad 132: *"...We appreciate how the team contacts us at our place and time of convenience—it is a big hospital, and we get lost trying to find locations, or we may be busy buying medicines..."*

This flexibility in implementation did, however, raise workload concerns, notably for clinicians.

Trainer 1: *"Sometimes it gets challenging when there are multiple recruits. It is a big hospital; patients move location, and coordinating with families to train them within working hours is difficult..."*

**Individual characteristics**: Some dyads declined due to the perceived burden of care associated with task shifting:

Dyad 76: *"He was absolutely fine, only earning member. We can't take him home. We will sell everything we own and wait here for him to get better..."*

Neuro-speciality clinicians generally supported the intervention:

TT1: *"This endeavour will improve the safety of discharged patients; we are hard-pressed for time to conduct optimal counselling, training or follow-ups, and bed pressure forces us to discharge many patients…seeing their photographs at home, it is satisfying to know there is now a safety net…"*

But other specialities, notably those working with trauma and acute medical admissions, raised concerns:

TT4: *"This is a public hospital—how can we ask families to leave when the patient is still bed-bound and has a tracheostomy? What about medicolegal concerns, especially in poisoning cases, if they deteriorate at home?"*

**Outer setting**: Socioeconomic constraints and fears of social stigma presented as significant barriers to all stakeholders and were most prevalent for those from the poorest and the wealthiest socioeconomic groups:

Dyad 112: *"You are asking us to go home… we stay under the highway bridge..in a make-shift tent"* [meaning 'home' was not a place to care for a dependent]

Dyad 184: *"Last time we had taken our grandmother home, she did not have a tracheostomy. But giving feed through a tube in the nose was also very difficult: we have no machines like a grinder or fridge. Going to the nearest health centre when she pulled out the nose-tube was a nightmare..."*

**Innovation characteristics**: Access to equipment and support was viewed positively:

Dyad 56: *"The machines were very helpful. We would have been very confused about what to buy... and where to put which equipment by making a map of the room where we plan to keep our father"*

The mHealth app was also valued.

Dyad 105: *"This [the m-health app with instant messaging and calls] is very helpful. Queries about loose stools and hiccups that were solved simply over the phone would have otherwise scared us. Transporting my bedridden father to the Primary Health Centre would have been difficult, not to mention they would have promptly asked us to go to the referral centre 90 km away, the moment they saw the tracheostomy"*

## Discussion

This mixed-methods formative evaluation demonstrates that a structured family-led discharge programme for tracheostomised ICU survivors can be implemented within a public tertiary hospital in a resource-limited setting, with high stakeholder acceptability and strong clinician adoption over time. By engaging stakeholders throughout design and implementation and allowing tailoring of training, the intervention and approach to implementation, built trust among clinicians and carers, which appeared central to its implementation and adoption. Families and patient–carer dyads consistently rated the intervention as more acceptable, appropriate, and feasible than healthcare workers and administrators, underscoring the importance of centring patient and carer perspectives in health service intervention planning.

### Primary findings

More than 95% of eligible dyads consented, and two-thirds achieved the target of three hands-on training sessions. Carer confidence increased across all domains, and quality-of-life measures for patients and carers improved during follow-up. Importantly, feasibility ratings were not influenced by carer education or employment, suggesting the intervention is accessible and feasible across diverse socioeconomic groups. While younger trauma patients were less likely to be recruited, largely due to family concerns about burden and prognosis, lower-education carers demonstrated the ability to gain confidence through training, highlighting the adaptability of the intervention. Practical support, such as access to the equipment bank and the ability to communicate with clinical teams for support post-hospital discharge, was fundamental to implementation success.

                                                    

Barriers spanned multiple CFIR domains. Individual-level challenges included denial or reluctance among carers of previously healthy younger patients and concerns among clinicians about medicolegal risks, particularly for trauma cases [31,32] Process and inner-setting barriers centred on hospital bed pressures, fragmented training opportunities, and supply-chain difficulties for equipment [33]. Outer-setting challenges to home discharge included financial constraints, competing interests from rehabilitation centres, and ill-fitting provider reimbursement mechanisms [34,35]. Poor internet connectivity further limited the use of the mHealth application in some regions, although this was not intervention specific [36].

Facilitators included trust in the treating team, flexible and repeatable training, availability of essential equipment through a rental–retrieval model, and the reassurance of follow-up visits and telephone support [37]. The mHealth application provided valued access to remote guidance, though many carers preferred WhatsApp for its familiarity when communicating directly with clinical and research teams [38]. The Technology Acceptance Model (TAM) [37], explains why existing apps like WhatsApp with a high perceived ease of use, established perception of trust by users and familiar interfaces may be preferred over bespoke mHealth apps [38].

Over time, adoption among ICU clinicians increased substantially. By the third year, most referrals were initiated by treating teams, and even families had begun to play an active role, reflecting stronger engagement and growing confidence in the intervention. Similar patterns have been observed in other ICU implementation studies. Previous investigators have reported progressive improvement in compliance with the ABCDE bundle as familiarity increased [39] while targeted implementation strategies—such as education, feedback, and local champions—were linked to greater uptake of the ABCDEF bundle [40]. As teams integrate new processes into everyday routines, interventions often evolve from novel initiatives into accepted elements of standard care [41], reflecting the progressive embedding of practices described by normalization process theory [42] .

Although the study was not powered for clinical effectiveness, the decrease in time to home discharge and improvements observed in quality-of-life and care burden measures of patients and carers, suggest the potential for clinically meaningful benefits that warrant a multi-centre study.

## Adaptations and scale-up

Beyond the specific adaptations to the intervention and implementation documented here, the formative evaluation highlighted key considerations for future implementation. Ethical and medicolegal concerns raised by clinicians suggested that neurotrauma, post-neurosurgical stroke, and hypoxic brain injury dyads are best suited for initial scale-up. Establishing clear physiologic criteria for early carer recruitment will enable more timely engagement and training, while triangulated training approaches- including peer-to-peer support, may reduce inefficiencies linked to staff workload and precipitated discharge. Early involvement of families, reinforced by structured counselling and tailored follow-up, appears essential to maintaining trust and facilitating safe transition of care tasks and location.

Providing free equipment during the first month of discharge was particularly important for uninsured families, helping bridge the transition to home while allowing time to identify sustainable long-term options. Reliable follow-up, whether in-person or remote, proved critical for troubleshooting and maintaining carer confidence. Together, these findings suggest that the AIR model is not only feasible in an LMIC tertiary hospital setting but also has potential for scale-up if supported by deliberate carer selection, consistent clinician engagement, and integration with existing health system and digital communication resources.

## Strengths and limitations

Key strengths include the concurrent conduct of feasibility evaluation and pilot implementation, allowing for real-time feedback to shape the intervention. Using the MRC framework ensured contextual relevance while supporting alignment with implementation science principles. Importantly, lessons learned included the need for continuous stakeholder engagement, adaptability, and systematic planning, which will inform the future multi-centre scale-up and evaluation.

Limitations include the single-centre design, potential selection bias of dyads for psychometric assessments during recruitment, delays in deploying the mHealth application, and the absence of longer-term follow-up data on readmissions and complications. Carer support components, such as psychological or social interventions, were not formally integrated, and the intervention's financial impacts (direct and indirect), an important metric in LMIC contexts, were not assessed. Addressing these gaps will be essential for the future summative evaluation.

## Conclusion

This formative evaluation demonstrates that the AIR intervention is feasible, acceptable, and adaptable for supporting the transition of tracheostomised, functionally dependent ICU survivors to family-led home care in India. Recruitment and intervention reach were strong, and adoption by treating teams increased over time, with nearly all referrals originating from clinicians or families by the third year.

Feasibility and acceptability were consistently rated highly by both carers and patients, with families perceiving the intervention more positively than healthcare workers. Importantly, carer education and employment status were not associated with feasibility or confidence scores, suggesting the intervention is accessible across diverse socioeconomic groups. Confidence in care delivery improved progressively with training, and two-thirds of dyads achieved the target of at least three skills sessions, despite challenges of bed pressures and precipitated discharges.

The mHealth application was widely adopted, with over 70% of dyads installing it, though WhatsApp remained the preferred platform for communication. The equipment bank supported more than half of families, particularly those without insurance, by providing essential devices during the critical early transition phase. Home follow-up was achieved for more than 90% of patients living within 200 km, with telephone calls and in-person visits highly valued for troubleshooting and reassurance.

These findings suggest that family-led discharge models may offer a scalable strategy to address ICU bed pressures and limited long-term care capacity in LMIC health systems. The lessons learned highlight priority adaptations for scale-up, including focusing initially on neurotrauma and post-neurosurgical populations, setting clear physiologic criteria for early recruitment, integrating peer-to-peer training, and strengthening follow-up systems. A multicentre summative evaluation is now underway using the MRC-recommended frameworks.

## Supporting information

**S1 Fig. Compliance with standardised training sessions across patient-caregiver dyads.** The figure reflects adherence to the training component of the intervention and reasons for non-compliance.
(PDF)

**S2 Fig. Structured reporting for post-discharge follow-up of home-discharged patients.**
(PDF)

**S1 Table. Adaptations to AIR Implementation.**
(DOCX)

**S2 Table. Outcome variables with corresponding assessment time points.**
(DOCX)

**S1 File. - Protocol for Summative Evaluation.**
(DOCX)

**S2 File. Interview Guide.**
(DOCX)

## Acknowledgments

We thank the patients, caregivers, and healthcare workers who participated in the study. Special thanks to the AIIMS Bhubaneswar administration, for their support. We also acknowledge the contributions of the AIR research and implementation teams.

## Author contributions

**Conceptualization:** Swagata Tripathy, Asha P. Shetty, Nanda Kumar P.

**Data curation:** Swagata Tripathy, Asha P. Shetty, Nanda Kumar P, Sujata Mahapatra.

**Formal analysis:** Swagata Tripathy.

**Funding acquisition:** Swagata Tripathy, Rashan Haniffa, Abi Beane.

**Investigation:** Swagata Tripathy, Upendra Hansda, Nanda Kumar P, Alok Kumar Sahoo, Mahalingam V, Sujata Mahapatra, Kasturi Sanyal, Itimayee Panda, Guruprasad N.

**Methodology:** Swagata Tripathy, Upendra Hansda, Mahalingam V, Itimayee Panda, Guruprasad N, Rashan Haniffa, Abi Beane.

**Project administration:** Swagata Tripathy, Upendra Hansda, Mahalingam V, Sujata Mahapatra, Jayanta Kumar Mitra, Parnandi Bhaskar Rao, Kasturi Sanyal, Guruprasad N, Jagannath Sahoo.

**Supervision:** Swagata Tripathy, Alok Kumar Sahoo, Jayanta Kumar Mitra, Parnandi Bhaskar Rao, Rashan Haniffa, Abi Beane.

**Validation:** Swagata Tripathy, Rashan Haniffa, Abi Beane.

**Visualization:** Swagata Tripathy, Rashan Haniffa, Abi Beane.

**Writing – original draft:** Swagata Tripathy, Nanda Kumar P, Abi Beane.

**Writing – review & editing:** Swagata Tripathy, Upendra Hansda, Kasturi Sanyal, Itimayee Panda, Rashan Haniffa, Abi Beane.

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
