## [Decision Letter · Decision Letter 0]

27 Feb 2026

PONE-D-25-65437Family-Led post-ICU Discharge Intervention for Tracheostomised Patients in India: Feasibility and Formative Impact Evaluation.PLOS OneDear Dr. Tripathy,

Thank you for submitting your manuscript to PLOS ONE. After careful consideration, we feel that it has merit but does not fully meet PLOS ONE’s publication criteria as it currently stands. Therefore, we invite you to submit a revised version of the manuscript that addresses the points raised during the review process.

The manuscript has been evaluated by three reviewers, and their comments are available below.

Could you please carefully revise the manuscript to address all comments raised?

We look forward to receiving your revised manuscript.

Kind regards,

Ilse Bloom

Staff Editor

PLOS One

“National Health Mission India, Wellcome Trust.”

4. Thank you for stating the following in the Funding Section of your manuscript:

“The study was funded by the National Health Mission, Odisha, and Wellcome Innovations Flagship Collaboration for Research Implementation and Training (224048/Z/21/Z). Institutional ethics committee approval was obtained, and the trial was prospectively registered (CTRI/2020/11/029443). Written informed consent was obtained from all participants.”

“National Health Mission India, Wellcome Trust.”

5. We note that you have indicated that there are restrictions to data sharing for this study. For studies involving human research participant data or other sensitive data, we encourage authors to share de-identified or anonymized data. However, when data cannot be publicly shared for ethical reasons, we allow authors to make their data sets available upon request. For information on unacceptable data access restrictions, please see http://journals.plos.org/plosone/s/data-availability#loc-unacceptable-data-access-restrictions.

Reviewers' comments:

Reviewer's Responses to Questions

**Comments to the Author**

1. Is the manuscript technically sound, and do the data support the conclusions?

Reviewer #1: Yes

Reviewer #2: Partly

Reviewer #3: Partly

2. Has the statistical analysis been performed appropriately and rigorously? 

Reviewer #1: Yes

Reviewer #2: N/A

Reviewer #3: No

3. Have the authors made all data underlying the findings in their manuscript fully available?

Reviewer #1: No

Reviewer #2: No

Reviewer #3: Yes

4. Is the manuscript presented in an intelligible fashion and written in standard English?

Reviewer #1: Yes

Reviewer #2: Yes

Reviewer #3: Yes

5. Review Comments to the Author

Reviewer #1: Dear Author,

This is a well-conducted and timely formative evaluation of a co-designed, multicomponent intervention aimed at supporting the transition of tracheostomised, functionally dependent ICU survivors to family-led home care in an LMIC setting. The study addresses an important gap in post-ICU transitional care, particularly in resource-constrained contexts. The integration of implementation science frameworks (CFIR and MRC guidance) strengthens the conceptual and methodological grounding of the work.

Overall, the manuscript is clearly written, methodologically sound, and presents meaningful findings with practical implications for scale-up. I recommend minor revision, primarily to enhance clarity, precision, and structural consistency.

My proposed revisions are as follows:

1. The manuscript reports improvements in patient and carer quality-of-life measures. Please clarify whether these improvements were: Statistically significant, clinically meaningful, or both. A brief statement on clinical relevance would strengthen interpretation.

2. While feasibility and acceptability are clearly central constructs, it would be helpful to explicitly restate the primary outcome(s) in the Discussion to reinforce alignment between study objectives and findings.

3. A few minor technical points: Ensure consistent formatting of references (e.g., spacing before citation numbers). Standardise terminology (e.g., “inner-setting” instead of “innersetting”). Correct minor punctuation inconsistencies (e.g., hyphen use in “MRC-recommended frameworks”).

4. The description of increasing clinician adoption over three years is compelling. Consider slightly strengthening the interpretive framing by explicitly linking this trajectory to known implementation theory (e.g., normalization of innovations into routine care).

5. The finding that many carers preferred WhatsApp over the purpose-built application is highly interesting. The Discussion could briefly expand on: Implications for digital health design in LMICs, whether leveraging existing communication platforms may enhance sustainability.

Reviewer #2: I enjoyed reading your paper and found it to be very interesting. In my opinion your paper handles about a very important topic with a set of interventions that is not only important to implement in MLIC, but also has a growing importance for families in high income countries. Nevertheless, the intervention under evaluation is rather complex, and clearly describing all that was undertaken is not straightforward. The authors have provided a generally clear description of the work performed, but the manuscript would benefit from additional revisions to further improve clarity and readability. I have a few comments and recommendations that you may consider that I believe will help you to clarify and strengthen your manuscript. I will try to order my comments according to the order of your manuscript.

Abstract

The abstract does not fully reflect the amount of work that has been carried out, nor the quality of the intervention the authors have developed and evaluated. With some adjustments, this can be improved. For instance, at several points it appears as though a word or part of a sentence is missing.

For example:

- ‘66% of carers completed ≥3 sessions, and 61% reached target competence after three.’ ….. After three??? Months? Sessions? Years? This is unclear.

- ‘More than half of the families? accessed equipment through the rental–retrieval bank, and 91% of eligible families received in-person follow-up.’

In addition, there are too many abbreviations in the methods section of the abstract (utilized measurement tools) that are not explained, which makes it difficult to interpret the results.

Introduction

Please describe some characteristics of the ICU where the intervention has taken place.

Methods

The bundle of interventions that the authors have implemented and evaluate in this paper is, in my view, very well designed and highly meaningful. Clear measurement methods have been used; however, I miss a description of the criteria for being selected to participate in the assessments and interviews. I understand that in implementation and developmental research it is common not to expose all participants to questionnaires, but it remains unclear why these particular participants were chosen.

Furthermore, there is no adequate description of all measurement instruments, and for some instruments no reference is provided (e.g., for the RE-AIM framework) nor is a citation included in the description of the respective instrument.

I also miss an example of the interview guide as well as information about the characteristics of the interviewers, the healthcare workers who delivered the training and the researchers. Such details are important for assessing the credibility and transferability of the qualitative findings. Especially because the authors describe that reflexive practice, including regular team meetings conducted with experienced qualitative researchers, ensured analytic credibility. Please elaborate about that expertise.

Results

The results are well described and clearly illustrate the barriers and facilitators to scaling up this intervention more broadly. The authors have also used illustrative and meaningful quotes.

Figures and tables

Regarding the TREND Statement Checklist: I could not find page numbers in the manuscript, which makes it difficult to locate where the listed items are addressed. I would recommend adding these page numbers.

Regarding the protocol that was added as a supplement, I would recommend adding the flowchart that outlines the entire project to the main manuscript, but in a much more readable format. This would greatly help the reader to understand the timelines of the interventions and the corresponding measurement points.

I personally find the flowcharts somewhat less elegantly presented, although they are clear in terms of content. The language used could be more consistent as well (e.g., using either “expired” or “deceased,” rather than switching between the two).

I wish the authors much success with revising the manuscript and with the further implementation of the intervention.

Reviewer #3: Greater clarity in reporting the assessment/evaluation time points would strengthen the manuscript. The statistical analysis approach is not clearly outlined, and incorporating inferential statistical analyses where appropriate could enhance the findings. Attention to consistent formatting and thorough proofreading would further improve the presentation of the manuscript.

Abstract: Abbreviations to be spelled out first, prior to usage.

At least 1 decimal place is required for the percentage figure. This includes the figures in the text, tables, etc.

Exclusion criteria are to be stated.

The sampling strategy for participant recruitment is unclear, particularly given the use of a convenience sampling approach. More details are to be provided.

Dyads are to be defined.

Page 19: The statement, ‘By year three, nearly all referrals were made directly by treating teams or families (64/65, 98.5%),’ lacks clarity. Please clarify how this figure was derived, including the definitions of the numerator and denominator.

Page 19: Supplementary table *? Supplementary Table 1 was not found in the manuscript.

Page 20: took ~1 hour (symbol ~ is to be replaced with approximately)

Page 20: For ‘Median time to discharge was 10 (SD 3.5)’, IQR is to be displayed rather than SD.

Page 20 & 21: Some words appear excessively spaced or stretched to accommodate page alignment and should be adjusted to ensure proper formatting and readability.

In Table 1, the decimal places for percentages (%) are to be standardised throughout. Female should be presented together with male under the same variable category. The table requires minor formatting adjustments for improved presentation. A separating row is to be included between variables to enhance clarity. Additionally, bold formatting should be avoided for subcategories/stems to ensure consistency.

In Table 2, the ‘definitions’ of “Range” and “IQR” are be clearly denoted in the table footnote. The font size is to be standardised throughout the table. For the healthcare workers’ data, the IQR value is to be clearly separated/spaced out from the Range value to improve readability.

It would strengthen the manuscript if the flow chart incorporated the assessment/evaluation time points.

While multiple time points were used to assess the outcome variables, these were not clearly presented, and inferential statistical analyses were not reported. Including a table summarising the assessment time points for each outcome variable would enhance clarity. Statistical analyses are to be aligned with the specified assessment time points where appropriate, and effect sizes with 95% confidence intervals could be presented where applicable.

Further details regarding the qualitative analysis process, including the analytical approach and software used, would also enhance transparency.

Description on missing data and method handling is to be provided (if any).

References did not conform to the journal format.

6. PLOS authors have the option to publish the peer review history of their article (what does this mean?). If published, this will include your full peer review and any attached files.

Reviewer #1: No

Reviewer #2: No

Reviewer #3: No

---

## [Author Response · Author response to Decision Letter 1]

26 Mar 2026

Title: Response to Reviewers – Manuscript PONE-D-25-65437

Manuscript: Family-Led post-ICU Discharge Intervention for Tracheostomised Patients in India: Feasibility and Formative Impact Evaluation

We thank the Editor and reviewers for their thoughtful and constructive feedback. We are encouraged that the reviewers found the study timely and methodologically sound. We have revised the manuscript to address all comments and have improved clarity in the description of the intervention, study design, measurement methods, and reporting of results. All changes have been incorporated into the revised manuscript and highlighted in the tracked-changes version.

Reviewer #1: Dear Author,

This is a well-conducted and timely formative evaluation of a co-designed, multicomponent intervention aimed at supporting the transition of tracheostomised, functionally dependent ICU survivors to family-led home care in an LMIC setting. The study addresses an important gap in post-ICU transitional care, particularly in resource-constrained contexts. The integration of implementation science frameworks (CFIR and MRC guidance) strengthens the conceptual and methodological grounding of the work.

Overall, the manuscript is clearly written, methodologically sound, and presents meaningful findings with practical implications for scale-up. I recommend minor revision, primarily to enhance clarity, precision, and structural consistency.

My proposed revisions are as follows:

1. The manuscript reports improvements in patient and carer quality-of-life measures. Please clarify whether these improvements were: Statistically significant, clinically meaningful, or both. A brief statement on clinical relevance would strengthen interpretation.

We thank the reviewer for this suggestion. We have clarified in the Results section -the observed improvements in patient and caregiver quality-of-life measures were statistically significant. We have added a brief statement discussing the clinical relevance of these changes in the Indian context; Page 13, Results. The improvement was significant between hospital discharge and day 14 (p < 0.001) and discharge and day 28 (p < 0.001), while the difference between day 14 and day 28 was not statistically significant (p = 0.06).

Page 17, Discussion. Although the study was not powered for clinical effectiveness , the decrease in time to home discharge and improvements observed in quality-of-life measures for both patients and carers, suggest the potential for clinically meaningful benefits that warrant a multi-centre study.

2. While feasibility and acceptability are clearly central constructs, it would be helpful to explicitly restate the primary outcome(s) in the Discussion to reinforce alignment between study objectives and findings.

Opening paragraph of the Discussion revised as follows.

Page 15, Discussion. This mixed-methods formative evaluation demonstrates that a structured family-led discharge programme for tracheostomised ICU survivors can be implemented within a public tertiary hospital in a resource-limited setting, with high stakeholder acceptability and strong clinician adoption over time.

3. A few minor technical points: Ensure consistent formatting of references (e.g., spacing before citation numbers). Standardise terminology (e.g., “inner-setting” instead of “innersetting”). Correct minor punctuation inconsistencies (e.g., hyphen use in “MRC-recommended frameworks”).

We thank the reviewer for noting these inconsistencies. The manuscript (and reference list) has been carefully edited to standardize terminology including for the above examples.

4. The description of increasing clinician adoption over three years is compelling. Consider slightly strengthening the interpretive framing by explicitly linking this trajectory to known implementation theory (e.g., normalization of innovations into routine care).

We appreciate this suggestion. We have strengthened the interpretive framing in the discussion as follows.

Page 16, Discussion. As teams integrate new processes into everyday routines, interventions often evolve from novel initiatives into accepted elements of standard care,(42) reflecting the progressive embedding of practices described by normalization process theory.(43).

5. The finding that many carers preferred WhatsApp over the purpose-built application is highly interesting. The Discussion could briefly expand on: Implications for digital health design in LMICs, whether leveraging existing communication platforms may enhance sustainability.

We agree and have expanded the Discussion as follows;

Page 16, Discussion. The Technology Acceptance Model (TAM) (37), explains why existing apps like WhatsApp with a high perceived ease of use, established perception of trust by users and familiar interfaces may be preferred over bespoke mHealth apps.(38)

Reviewer #2: I enjoyed reading your paper and found it to be very interesting. In my opinion your paper handles about a very important topic with a set of interventions that is not only important to implement in MLIC, but also has a growing importance for families in high income countries. Nevertheless, the intervention under evaluation is rather complex, and clearly describing all that was undertaken is not straightforward. The authors have provided a generally clear description of the work performed, but the manuscript would benefit from additional revisions to further improve clarity and readability. I have a few comments and recommendations that you may consider that I believe will help you to clarify and strengthen your manuscript. I will try to order my comments according to the order of your manuscript.

Abstract

The abstract does not fully reflect the amount of work that has been carried out, nor the quality of the intervention the authors have developed and evaluated. With some adjustments, this can be improved. For instance, at several points it appears as though a word or part of a sentence is missing.

For example:

- ‘66% of carers completed ≥3 sessions, and 61% reached target competence after three.’ ….. After three??? Months? Sessions? Years? This is unclear.

- ‘More than half of the families? accessed equipment through the rental–retrieval bank, and 91% of eligible families received in-person follow-up.’

In addition, there are too many abbreviations in the methods section of the abstract (utilized measurement tools) that are not explained, which makes it difficult to interpret the results.

We thank the reviewer for identifying areas for improvement in the abstract we have corrected all these concerns now.

Introduction

Please describe some characteristics of the ICU where the intervention has taken place.

We have expanded the description of the study setting as follows;

Page 9, Setting. The study was conducted in the All-India Institute of Medical Sciences (AIIMS) Bhubaneswar, a tertiary care teaching hospital in Odisha India. AIIMS has approximately 100 ICU beds across multiple units including central (mixed medical-surgical) ICU, neuro-respiratory ICU, neurosurgical ICU, and emergency ICUs. Patients for the intervention were recruited from the neurology, neurosurgical and mixed medical–surgical ICUs. These ICUs have a total capacity of 35 beds and manage critically ill patients requiring advanced monitoring and organ support. The units function with a patient-to-nurse ratio of approximately 1:2–3, facilitating continuous bedside monitoring and intensive nursing care. Clinical management is supervised by trained intensivists who provide 24-hour on-call coverage, along with full-time in-person medical staff available in the unit at all times.

Methods

The bundle of interventions that the authors have implemented and evaluate in this paper is, in my view, very well designed and highly meaningful. Clear measurement methods have been used; however, I miss a description of the criteria for being selected to participate in the assessments and interviews. I understand that in implementation and developmental research it is common not to expose all participants to questionnaires, but it remains unclear why these particular participants were chosen.

We have clarified the participant selection process for quantitative assessments and qualitative interviews as follows;

Page 10, Eligibility and recruitment. While all eligible patient-carer dyads and stakeholders were offered participation in the intervention, convenience sampling was used for psychometric assessments, and purposive sampling was used for qualitative interviews to ensure representation of different stakeholder groups and experiences with the intervention.

Furthermore, there is no adequate description of all measurement instruments, and for some instruments no reference is provided (e.g., for the RE-AIM framework) nor is a citation included in the description of the respective instrument.

Thank you for the suggestion. We have expanded the Methods section as follows;

Page 10-11 : Implementation frameworks and outcome measures The Medical Research Council (MRC) framework for developing and evaluating complex healthcare interventions guided the overall design and formative evaluation of the intervention, including assessment of feasibility, implementation processes, and contextual influences on delivery.(22) To assess implementation performance across multiple domains, we applied the RE-AIM framework (Reach, Effectiveness, Adoption, Implementation, Maintenance), which provides a structured approach for evaluating public health interventions across individual and organisational levels.(17) Both are widely used and accepted guides for implementation research internationally. Stakeholder perceptions of the intervention were assessed using validated implementation outcome measures: the Acceptability of Intervention Measure (AIM), Intervention Appropriateness Measure (IAM), and Feasibility of Intervention Measure (FIM).(23) Additional quantitative measures included carer confidence scores related to tracheostomy and home-care skills, as well as patient and caregiver outcome measures including quality-of-life (24)and caregiver burden assessments.(25) To explore contextual factors influencing implementation, semi-structured qualitative interviews were conducted with carers and healthcare workers. Qualitative data were analysed using the Consolidated Framework for Implementation Research (CFIR), which provides a structured taxonomy for identifying barriers and facilitators across domains including intervention characteristics, inner and outer settings, individual characteristics, and implementation processes; and is again widely accepted and used in international implementation research.(19)

I also miss an example of the interview guide as well as information about the characteristics of the interviewers, the healthcare workers who delivered the training and the researchers. Such details are important for assessing the credibility and transferability of the qualitative findings. Especially because the authors describe that reflexive practice, including regular team meetings conducted with experienced qualitative researchers, ensured analytic credibility. Please elaborate about that expertise.

We thank the reviewer for this suggestion and have updated the section as follows; ,

Page 12- Data Collection and Analyses- The interview guide for the qualitative assessment was prepared by members of the research team and can be found in Supplement 7. Interviews were conducted by project team members familiar with the AIR programme, trained in qualitative methods, but not involved in delivering the intervention. Field notes and interview transcripts were reviewed by members of the research team and analysed using an interpretive approach. Analysis was carried out alongside ongoing data collection, allowing emerging issues to be explored in later interviews. Senior researchers with a track record in critical care service improvement and implementation facilitated the reflexive sessions. Emergent findings were discussed with both senior research and the wider multidisciplinary AIR team to reflect on interpretation and incorporate perspectives from stakeholders.

Results

The results are well described and clearly illustrate the barriers and facilitators to scaling up this intervention more broadly. The authors have also used illustrative and meaningful quotes.

Figures and tables

Regarding the TREND Statement Checklist: I could not find page numbers in the manuscript, which makes it difficult to locate where the listed items are addressed. I would recommend adding these page numbers.

We have revised the TREND checklist to include page numbers.

Regarding the protocol that was added as a supplement, I would recommend adding the flowchart that outlines the entire project to the main manuscript, but in a much more readable format. This would greatly help the reader to understand the timelines of the interventions and the corresponding measurement points.

We have now added a flowchart outlining the intervention and process evaluation as Figure 1.

I personally find the flowcharts somewhat less elegantly presented, although they are clear in terms of content. The language used could be more consistent as well (e.g., using either “expired” or “deceased,” rather than switching between the two).

We have standardized terminology throughout the manuscript and figures (e.g., using “deceased” consistently rather than alternating terms).

I wish the authors much success with revising the manuscript and with the further implementation of the intervention.

Reviewer #3: Greater clarity in reporting the assessment/evaluation time points would strengthen the manuscript. The statistical analysis approach is not clearly outlined, and incorporating inferential statistical analyses where appropriate could enhance the findings. Attention to consistent formatting and thorough proofreading would further improve the presentation of the manuscript.

We have now included a new Figure (1) based on a similar suggestion by Reviewer 2, that better illustrates the assessment time points.

Data Analyses Page 12. Quantitative data were analysed using descriptive statistics. Continuous variables were summarised using means with standard deviations (SD) or medians with interquartile ranges (IQR), depending on distribution, and categorical variables were summarised as frequencies and percentages. Changes in patient quality of life (EQ-5D) and caregiver burden (CBS) across time points were assessed using non-parametric tests (Mann–Whitney U or Wilcoxon rank-sum tests) as appropriate due to non-normal distributions. Effect sizes were estimated using rank-based measures where relevant, and 95% confidence intervals were reported where applicable. All analyses were exploratory and intended to inform future hypothesis-testing studies.

Abstract: Abbreviations to be spelled out first, prior to usage. This has been corrected.

At least 1 decimal place is required for the percentage figure. This includes the figures in the text, tables, etc. This has been corrected.

Exclusion criteria are to be stated.

We have included the criteria.

Page 10- Patients were excluded if they were <18 years of age, decannulated prior to discharge, or lacked an available caregiver able to participate in training.

The sampling strategy for participant recruitment is unclear, particularly given the use of a convenience sampling approach. More details are to be provided.

Sampling strategy has now been clarified in Methods (Eligibility and Recruitment section) as also suggested by Reviewer 2.

Page 10, Eligibility and recruitment. While all eligible patient-carer dyads and stakeholders were offered participation in the intervention, convenience sampling was used for psychometric assessments, and purposive sampling was used for qualitative interviews to ensure representation of different stakeholder groups and experiences with the intervention.

Dyads are to be defined.

The definition has been added

Page 10- A dyad refers to the patient and their pri

---

## [Decision Letter · Decision Letter 1]

15 Apr 2026

Family-Led post-ICU Discharge Intervention for Tracheostomised Patients in India: Feasibility and Formative Impact Evaluation.

PONE-D-25-65437R1

Dear Dr. Tripathy,

We’re pleased to inform you that your manuscript has been judged scientifically suitable for publication and will be formally accepted for publication once it meets all outstanding technical requirements.

Kind regards,

Marianne Clemence

Staff Editor

PLOS One

Additional Editor Comments (optional):

Reviewers' comments:

Reviewer's Responses to Questions

**Comments to the Author**

1. If the authors have adequately addressed your comments raised in a previous round of review and you feel that this manuscript is now acceptable for publication, you may indicate that here to bypass the “Comments to the Author” section, enter your conflict of interest statement in the “Confidential to Editor” section, and submit your "Accept" recommendation.

Reviewer #1: All comments have been addressed

Reviewer #3: All comments have been addressed

2. Is the manuscript technically sound, and do the data support the conclusions?

Reviewer #1: Yes

Reviewer #3: (No Response)

3. Has the statistical analysis been performed appropriately and rigorously? 

Reviewer #1: Yes

Reviewer #3: (No Response)

4. Have the authors made all data underlying the findings in their manuscript fully available?

Reviewer #1: Yes

Reviewer #3: (No Response)

5. Is the manuscript presented in an intelligible fashion and written in standard English?

Reviewer #1: Yes

Reviewer #3: (No Response)

6. Review Comments to the Author

Reviewer #1: The authors have made the corrections I suggested. Therefore, my decision will be to accept the article.

Reviewer #3: (No Response)

7. PLOS authors have the option to publish the peer review history of their article (what does this mean?). If published, this will include your full peer review and any attached files.

Reviewer #1: No

Reviewer #3: No

---

## [Editor Report · Acceptance letter]

PONE-D-25-65437R1

PLOS One

Dear Dr. Tripathy,

I'm pleased to inform you that your manuscript has been deemed suitable for publication in PLOS One. Congratulations! Your manuscript is now being handed over to our production team.

Kind regards,

on behalf of

Dr Marianne Clemence

Staff Editor

PLOS One